# AFR-BERT: Attention-based mechanism feature relevance fusion multimodal sentiment analysis model

**Ji Mingyu**[☯]*, **Zhou Jiawei**[iD][☯]*, **Wei Ning**

Department of Software Engineering, Faculty of Information and Computer Engineering, The Northeast Forestry University, Harbin, China

☯ These authors contributed equally to this work.
* jimingyu@nefu.edu.cn (JM); zzjw@nefu.edu.cn (ZJ)

**Data Availability Statement:** All relevant data are within the manuscript and its Supporting information files.

## Abstract

Multimodal sentiment analysis is an essential task in natural language processing which refers to the fact that machines can analyze and recognize emotions through logical reasoning and mathematical operations after learning multimodal emotional features. For the problem of how to consider the effective fusion of multimodal data and the relevance of multimodal data in multimodal sentiment analysis, we propose an attention-based mechanism feature relevance fusion multimodal sentiment analysis model (AFR-BERT). In the data pre-processing stage, text features are extracted using the pre-trained language model BERT (Bi-directional Encoder Representation from Transformers), and the BiLSTM (Bi-directional Long Short-Term Memory) is used to obtain the internal information of the audio. In the data fusion phase, the multimodal data fusion network effectively fuses multimodal features through the interaction of text and audio information. During the data analysis phase, the multimodal data association network analyzes the data by exploring the correlation of fused information between text and audio. In the data output phase, the model outputs the results of multimodal sentiment analysis. We conducted extensive comparative experiments on the publicly available sentiment analysis datasets CMU-MOSI and CMU-MOSEI. The experimental results show that AFR-BERT improves on the classical multimodal sentiment analysis model in terms of relevant performance metrics. In addition, ablation experiments and example analysis show that the multimodal data analysis network in AFR-BERT can effectively capture and analyze the sentiment features in text and audio.

## 1. Introduction

With the development of internet technology, people often express their feelings about their daily lives and opinions on hot topics, are used to reading product reviews before consumption, and like to write down their feelings after experiencing products and enjoying services. This has led to the generation of a huge amount of multimodal data. Mining sentiment in

**Funding:** This work is sponsored by National Natural Science Foundation of China (Grant No. 435 61901103).

**Competing interests:** The authors have declared that no competing interests exist.

these multimodal data has been a popular research topic in the fields of natural language processing, data mining, and user requirement analysis.

Traditional sentiment analysis studies are limited to a single textual modality that expresses sentiment primarily by words, phrases, and relationships [1]. Kim et al. [2] used convolutional neural networks to model sentences with convolution and pooling. Wang et al. [3] proposed a disconnected recurrent neural network that restricts the flow of textual information to a fixed number of steps. Lin et al. [4] used the self-attention mechanism in the sentence modeling process. According to numerous studies, it is not possible to effectively determine a person's emotional change based on a specific entity or event in front of us [5]. For sentiment analysis, a single modality cannot accurately determine a person's emotion. Psychologist Mehrabian [6] found that words only reflect 7% of emotions, voice and its characteristics reflect 38% of emotions, and expressions and body language reflect 55% of emotions in daily conversations, which indicates that facial expressions and voice convey the primary emotional information. However, since the emotion analysis of human micro-expressions and micro-motions is still not perfect, our research focuses mainly on text and audio. As the study progressed, the researchers discovered correlations and complementarities between the semantic information contained in the text and the phonetic information contained in the speech. Likewise, the researchers found that more comprehensive information could be provided for sentiment analysis through the emotional interaction between text and audio patterns [7]. How to effectively fuse text and audio information becomes one of the issues that must be considered in multimodal sentiment analysis.

To solve the above problems, we proposed an attention-based mechanism feature relevance fusion multimodal sentiment analysis model (AFR-BERT) based on the study in literature [8, 9]. We added audio data to the traditional text sentiment analysis, making the data itself more malleable and recognizable. First, text and audio features were pre-processed using a bi-directional encoder by transformers [10] (BERT [11]) and a bi-directional long short-term memory network (BiLSTM) [12] to obtain unimodal features of text and audio. After obtaining the unimodal features of text and audio, the multimodal features were fused using a multimodal data fusion network based on the attention mechanism. Then, the self-attention mechanism is adopted to reduce the dependence on external information and capture the internal correlation of features. Finally, the processed multimodal information was classified according to sentiments. To demonstrate the effectiveness of the method in this paper, we conducted researches and experiments on the public sentiment benchmark datasets CMU-MOSI [13] and CMU-MOSEI [14]. The experimental results show that our proposed multimodal sentiment analysis model can not only effectively fuse multimodal features and improve the accuracy of sentiment analysis, but also has the advantage of focusing on the relevance of model information.

## 2. Related work

**Multimodal sentiment analysis** has become an important research topic in natural language processing, mainly involving the computational study of information such as opinions and emotional states in data composed of text, images or audio, or even video [15]. Current research directions are focused on feature learning and multimodal fusion. For feature learning, Zadeh et al. [16] designed a multi-attentive memory fusion network for sentiment evaluation through view interaction. Hazarika et al. [17] proposed a multimodal sentiment analysis model with a fusion of text and audio feature learning and provided a self-focused mechanism for multimodal sentiment weighting. In contrast to the above feature fusion methods based on attention mechanisms, Liu et al. [18] proposed a low-rank multimodal fusion method that

uses a low-rank tensor to enhance the efficiency of the model and improve the performance of sentiment analysis while reducing the parameters. Hazarika et al. [19] proposed a modality-invariant and -specific sepresentations method, which incorporates a combination of losses including distributional similarity, orthogonal loss, reconstruction loss, and task prediction loss to learn modality-invariant and modality-specific representation. Concerning multimodal fusion, many researchers have proposed many solutions to proactively fuse multimodal features. So far, the main fusion strategies are feature fusion, decision fusion, and model fusion [20]. Feature fusion is the cascading of different unimodal features into multimodal features. Decision fusion is mainly a different combination strategy for different unimodal data. Model fusion methods combine feature fusion and decision fusion, which increases the complexity and training difficulty of the model while combining the advantages of both [21]. Priyasad et al. [22] blended text and audio information for sentiment analysis and designed a deep convolutional neural network (DCNN [23]) and recurrent neural network (RNN [24]) cascaded network to extract text and audio features, and finally fused different patterns of features by cross-attention layer to achieve sentiment analysis. D. Krishna et al. [8] proposed a cross-modal attention mechanism and a one-dimensional convolutional neural network to implement multimodal assignment and sentiment analysis with a 1.9% improvement in accuracy compared to previous methods. Poria et al. [25] proposed a deep learning model based on contextual BiLSTM [12], which uses the contextual information of audio data to obtain more sentiment features, independently examines and classifies the features of text modality and audio modality in decision fusion, and later fuses the results into a decision vector to develop a combined strategy. Although the above models investigated feature extraction and multimodal fusion methods, they all ignored the correlation and complementarity between semantic and speech information, which directly impact feature fusion and sentiment analysis results.

## Pre-trained language models

Influenced by transfer learning, pre-trained language models have made a breakthrough in natural language processing(NLP), learning generic language representations from massive corpora and greatly improving downstream tasks without the need for manual annotation. Peters et al. [26] proposed a novel deep bidirectional language model ELMo (Embeddings from Language Models). ELMo is a deep contextual word representation method. It is pre-trained on a large text corpus to obtain the generic semantic representations, and the obtained semantic representations are used as features to migrate to downstream tasks. After extensive experiments, ELMo has been shown to significantly improve the performance of NLP tasks. Radford et al. [27] proposed GPT(Generative Pre-Training), a generative pre-trained transformer language model. GPT pre-trains word vectors on a large-scale unsupervised text corpus and fine-tunes them using small-scale supervised text data. Experiments show that GPT achieves surprising results in tasks such as text translation, semantic matching, knowledge quizzes, and inference. Devlin et al. [11] proposed the BERT based on the encoder bidirectional encoding representation in transformers [10]. BERT uses the new masked language model for pre-training to generate deep bi-directional language representations. The experiments show that BERT significantly outperforms other pre-trained language models with the latest results achieved in 11 NLP tasks.

**Multitask learning** [28] is a widely used learning method and is likewise a derived transfer learning method. The purpose of multitask learning is to optimize multiple learning tasks simultaneously and to improve the model's generalization and prediction performance for each task using shared information between tasks. Multitask learning is generally divided into two types. One type has a primary task and a secondary task. While the primary task is the

main function of the model, the secondary task is to help train the primary task. The other type has multiple equal tasks that build on each other. The first type is widely used in deep learning, where the selection of appropriate auxiliary tasks is crucial for the success of a multi-task learning framework [29]. The domain information of the training signal through related tasks in the main task is used as a derivation bias to improve the generalization of the main task. In recent years, multitask learning has come to be used in deep learning. Multitask learning allows deep optimization of models, enabling better access to data representations and more comprehensive and variable mining of data information. Yu et al. [30] focused on refining visual features by learning multiple related tasks simultaneously in given target information and proposed a goal-oriented multimodal BERT (TomBERT). The motivation is the observation that correlated images in the sample highlight the focused target and reflects the users' emotions towards the focused target. To be specific, TomBERT first learns the image features associated with the target and then uses the transformer model to fuse them with text features. Xu et al. [31] proposed the multiple interaction memory network, which includes two interaction memory networks for monitoring textual and visual information of a given target, to capture the global information of the data more comprehensively. The above studies show that the combination of multitask learning can help the model achieve better performance.

## 3. Materials and methods

### 3.1 Dataset

**CMU-MOSI** (Multimodal Opinion Level Sentiment Intensity) [13] is one of the most popular benchmark datasets, containing 93 videos with a total of 2199 conversations. Each conversation has a sentiment label in the range [-3,+3], and we define labels >0 as positive sentiment and labels < = 0 as negative sentiment. The training, validation, and test sets in CMU-MOSI each has 52, 10, and 31 videos with 1284 (679 positive conversations, 605 negative conversations), 229 (124 positive conversations, 105 negative conversations), and 686 (277 positive conversations, 409 negative conversations) conversations each. The division of our experimental dataset strictly follows the CMU-MOSI dataset format. Information about the CMU-MOSI dataset is shown in Table 1.

 **CMU-MOSEI**(Multimodal Opinion Sentiment and Emotion Intensity) [14] dataset comes from over 1,000 online YouTube speakers and contains 3229 videos with a total of 22676 conversations. Each conversation has an emotion label and these labels are divided into the range [-3,+3] at a coarse granularity. We define labels >0 as positive emotions and labels < = 0 as negative emotions, at fine granularity they are divided into 7 emotion labels: anger, disgust, fear, happiness, sadness, and surprise. The CMU-MOSEI training set, validation set, and test set each has 2550, 300, and 679 videos with 16216 (114992 positive conversations, 5219 negative conversations), 1835 (1333 positive conversations, 502 negative conversations), and 4625

**Table 1. CMU-MOSI dataset information.**

| CMU-MOSI | | | | |
|---|---|---|---|---|
| **Data** | **Train sets** | **Valid sets** | **Test sets** | **Total** |
| Video | 52 | 10 | 31 | 93 |
| Dialogue | 1284 | 229 | 686 | 2199 |
| Positive | 679 | 124 | 277 | 1080 |
| Negative | 605 | 105 | 409 | 1119 |
| Polarity | [-3,+3] | | | |

**Table 2. CMU-MOSEI dataset information.**

| | CMU-MOSEI | | | |
|---|---|---|---|---|
| Data | Train sets | Valid sets | Test sets | Total |
| Video | 2550 | 300 | 679 | 3229 |
| Dialogue | 16216 | 1835 | 4625 | 22676 |
| Positive | 11499 | 1333 | 3281 | 13213 |
| Negative | 4717 | 502 | 1344 | 6563 |
| Polarity | [-3,+3] | | | |

(3281 positive conversations, 1344 negative conversations) conversations. The relevant information of the CMU-MOSEI dataset is shown in Table 2.

### 3.2 Evaluation metrics

The performance evaluation metrics of the experiment include 2-class Accuracy ($ACC_2$), 7-class Accuracy ($ACC_7$), weighted average F1-score (F1), and mean absolute error (MAE).

The formula for calculating the accuracy rate is as follows:

$$Acc = (TP + TN)/(TP + FP + FN + TN) \tag{1}$$

The F1 is a weighted summed average of the precision and recall rates, and the metric is calculated as follows:

$$F1 = 2 \cdot \frac{precision \cdot recall}{precision + recall} \tag{2}$$

MAE is the absolute error between the predicted value and the true value. $\hat{y}_i$ denotes the true value and $y_i$ denotes the predicted value. The metric is calculated as follows:

$$MAE = \frac{1}{n}\sum_{1}^{n}|\hat{y}_i - y_i| \tag{3}$$

To demonstrate the validity of the model, the Pearson correlation coefficient (Corr) is used to further measure the degree of correlation between the predicted and true labels of the model. The closer Corr is to 1, the better the performance of the model. $\hat{y}, y, \bar{\hat{y}}, \bar{y}$ denotes the true and predicted values and their corresponding mean values, respectively. The metric is calculated as follows:

$$\rho_{\hat{y},y} = \frac{cov(\hat{y},y)}{\sigma_{\hat{y}}\sigma_y} = \frac{\sum_{i=1}^{n}(\hat{y}_i - \bar{\hat{y}})(y_i - \bar{y})}{\sqrt{\sum_{i=1}^{n}(\hat{y}_i - \bar{\hat{y}})^2 \times \sum_{i=1}^{n}(y_i - \bar{y})^2}} \tag{4}$$

### 3.3 Baseline

In the field of multimodal sentiment analysis, the classical models LMF, MFN, etc., and the recently proposed multimodal sentiment analysis models CM-BERT, Self-MM, etc. have achieved notable results. On the CMU-MOSI and CMU-MOSEI, AFR-BERT will perform performance comparison experiments with the baseline models. The baseline models are as follows:

**TFN.** The tensor fusion network uses a tensor fusion multimodal fusion approach to model intermodal dynamics, which aggregates single-, two-, and three-peak interactions [32].

**LMF.** The low-rank multimodal fusion network utilizes a low-rank weight tensor to improve multimodal fusion efficiency without compromising performance [4].

**MFN.** The memory fusion network adopts a neural network model with multi-view sequential learning for multimodal sentiment analysis by integrating view-specific information and cross-view information [14].

**RAVEN.** The recurrent attended variation embedding network model considers the fine-grained structure of nonverbal word sequences and dynamically changes the representation of words based on nonverbal cues [33].

**ICCN.** The interaction canonical correlation network exploits the outer product of feature pairs and typical deep learning analysis methods to study useful multimodal embedding features [34].

**MFM.** The multimodal factorization model generates discriminative targets by optimizing the cross-modal data and labels jointly and then ensures that the learned representations are rich in intra- and inter-modal features by differentiating the targets to predict label sentiment by distinguishing targets [35].

**MulT.** The multimodal transformer model is an end-to-end model that extends the standard transformer network to learn representations directly from unaligned multi-modal flows [36].

**CM-BERT.** The cross-modal BERT model introduces information from the audio modality to help the text modality fine-tune the pre-trained BERT model, and then uses a novel multi-modal attention fusion method that dynamically adjusts the word weights through the interaction of text and audio modalities [9].

**MISA.** The modality-invariant and -specific representations model learns a factorized subspace of each modality to provide a better representation as input to the fusion [19].

**Self-MM.** The self-supervised multitask multimodal model uses a self-supervised multitask learning strategy to improve pattern recognition accuracy by adjusting the weight of each subtask based on the design of multimodal labels and modal representation of the single-peak label generation module adjustment strategy [37].

**MAG-BERT.** The multimodal adaptation gate for BERT uses a gate structure connected to the BERT model to continuously improve the multimodal recognition accuracy of the model by modifying the BERT model with attention and adaptive vectors conditional on non-verbal behavior [38].

## 3.4 Method

Fig 1 presents the structure of the attention-based mechanism feature relevance fusion multimodal sentiment analysis model (AFR-BERT).

The AFR-BERT model consists of the following four main components.

1. **Data Preprocessing Layer**.

2. **Multimodal Fusion Layer**.

3. **Multimodal Association Layer**.

4. **Output Layer**.

   **The Data Preprocessing Layer** preprocesses text and audio data.

   **Text data.** The text data can be considered as consisting of phrases and relations with contextual relationships. The BERT is used in the experiments to obtain the output of the last layer

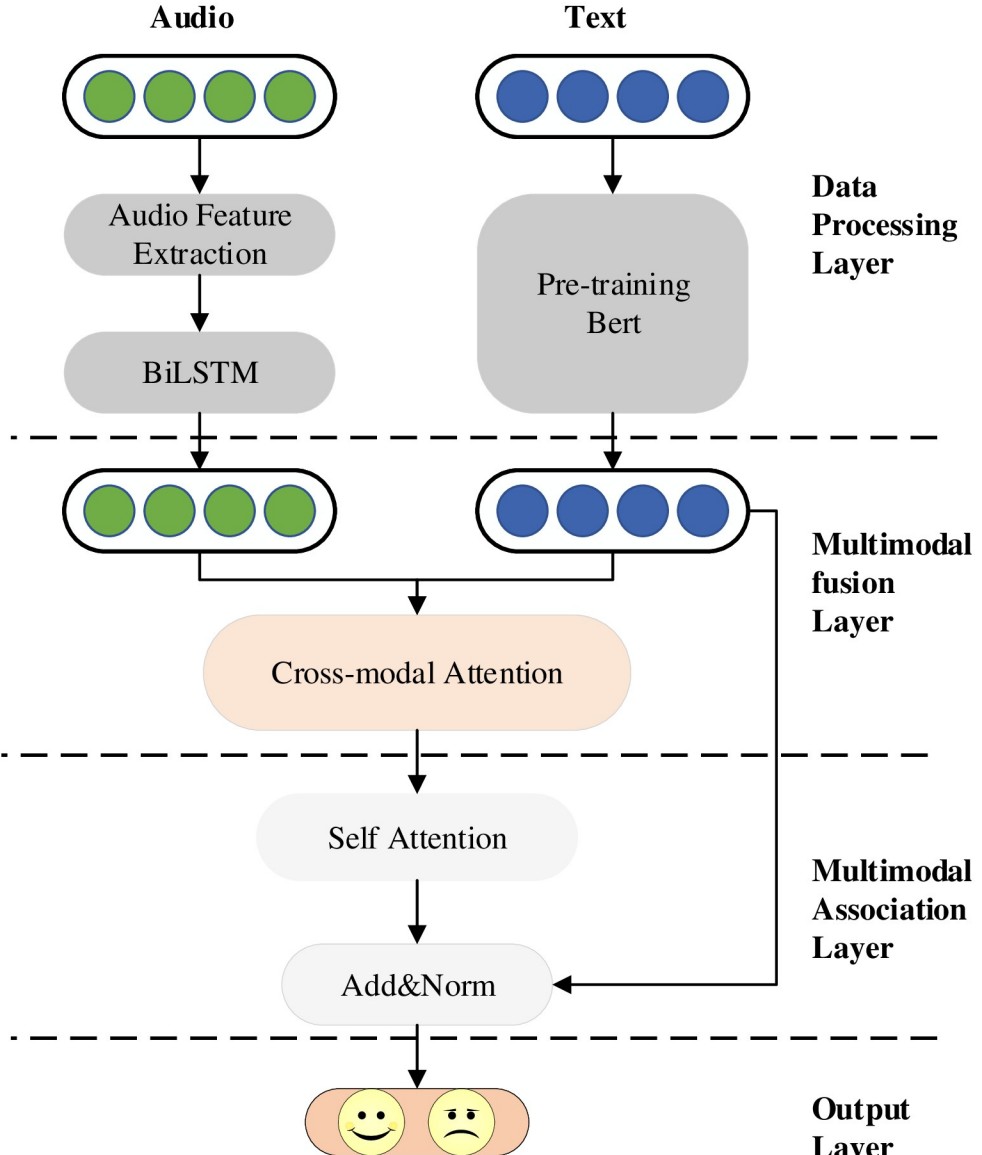

**Fig 1. Structure of AFR-BERT multimodal sentiment analysis model.** AFR-BERT is divided into four network modules, which correspond to data input, data fusion, data analysis, and data output.

of the encoder as text features. The text sequence for each word-piece token is T = [T$_1$, T$_2$, . . ., T$_n$], where n is the number of sequences length. The BERT model adds a CLS start classification identifier to the input sequence, and the output sequence is calculated after embedding and encoding as follows:

$$R_t = [X_{[CLS]}, X_1, X_2, \ldots .X_n] \tag{5}$$

**Audio data.** The speech signal is extracted by the COVAREP [39]. The time step of each word in the text data is obtained by the P2FA [40] so that the audio features are averaged over the corresponding step. Since multimodal feature fusion requires matrix operations on the data to ensure the same length as the text features, the null missing parts of the speech features

are filled with zeros. $A_{[CLS]}$ is the zero vector and the audio features are represented as below:

$$R_a = [A_{[CLS]}, A_1, A_2, \ldots A_n] \tag{6}$$

The audio feature representation with contextual information is obtained by BiLSTM [12]. LSTM [41] is a mechanism that uses memory cells and gates. It not only remembers long-term historical information but also solves the problem of gradient disappearance and gradient explosion. The LSTM core structure consists of forgetting gates, input gates, output gates, and memory cells. The expression of LSTM is calculated as below:

$$f_t = \sigma(\mathbf{W}_f \cdot [\mathbf{h}_{t-1}, x_t] + b_f) \tag{7}$$

$$i_t = (\mathbf{W}_i \cdot [\mathbf{h}_{t-1}, x_t] + b_i) \tag{8}$$

$$\tilde{\mathbf{C}}_t = \tanh(\mathbf{W}_c \cdot [\mathbf{h}_{t-1}, x_t] + b_c) \tag{9}$$

$$\mathbf{C}_t = f_t * \mathbf{C}_{t-1} + i_t * \tilde{\mathbf{C}}_t \tag{10}$$

$$\mathbf{o}_t = \sigma(\mathbf{W}_o \cdot [\mathbf{h}_{t-1}t, x_t] + b_o) \tag{11}$$

$$h_t = \mathbf{o}_t * \tanh(\mathbf{C}_t) \tag{12}$$

$x_t$ is the input feature of the audio at moment t. $C_t$ is the cell state. $\tilde{C}_t$ is the temporary cell state. $h_t$ is the hidden layer state of the audio t. $\sigma$ and *tanh* both are the activation functions. **W** is the weight matrix. b is the bias vector. $h_{t-1}$ is the hidden layer state at the previous moment. $f_t$ represents the forgetting gate. $i_t$ indicates the memory gate. $o_t$ means the output gate. Since one-way LSTM cannot utilize interdiscourse contextual information, Huang Z et al. [12] proposed BiLSTM (Bidirectional Long-Short Term Memory) to obtain long-term historical

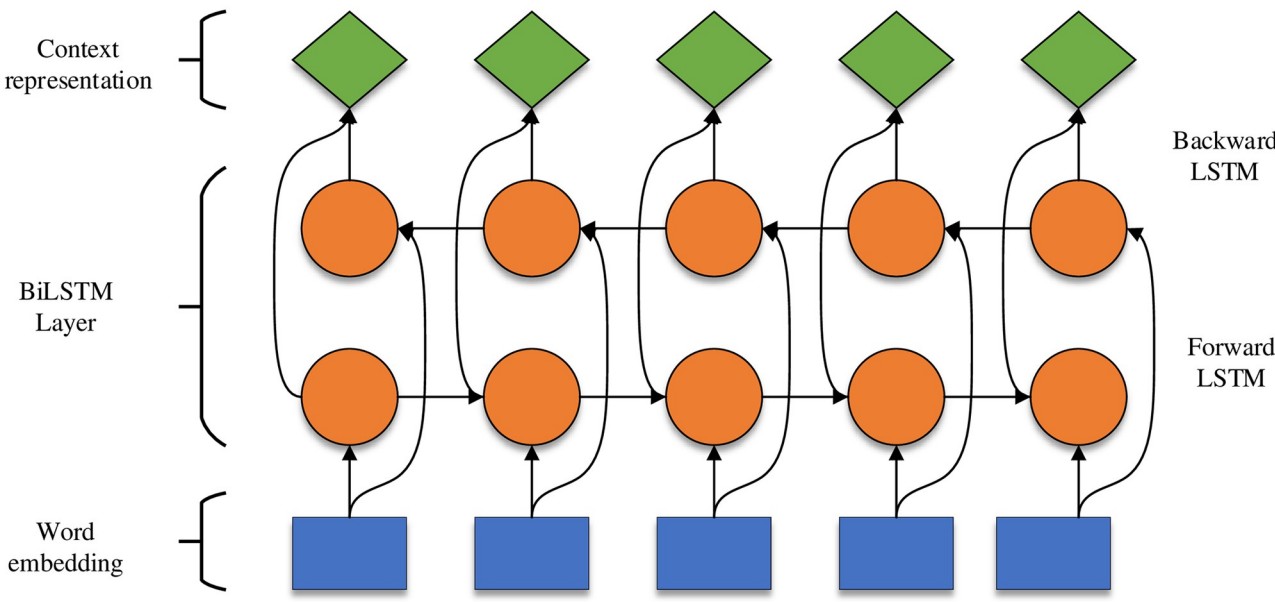

**Fig 2. BiLSTM model structure.** (Forward) means forward propagation of the model. (Backward) means model backward propagation.

information in each moment of discourse by forward and reverse LSTM. The specific structure is shown in Fig 2, with the following expressions:

$$\overrightarrow{\mathbf{h}_t} = \overrightarrow{\text{LSTM}}(x_t, \overrightarrow{\mathbf{h}_{t-1}}) \tag{13}$$

$$\overleftarrow{\mathbf{h}_t} = \overleftarrow{\text{LSTM}}(x_t, \overleftarrow{h_{t-1}}) \tag{14}$$

$$\mathbf{h}_t = [\overrightarrow{\mathbf{h}_t} \oplus \overleftarrow{\mathbf{h}_t}] \tag{15}$$

$\overrightarrow{\mathbf{h}_t}$ indicates LSTM forward output. $\overleftarrow{\mathbf{h}_t}$ indicates LSTM backward output. $[\oplus]$ is splicing operation. $h_t$ indicates BiLSTM output.

**The Multimodal Fusion Layer** fuses multimodal data features.

Fig 3 shows the structure of the cross-modal fusion attention mechanism proposed in this paper.

In this paper, multimodal sequence data include two main forms: text (T) and audio (A). Different extraction methods of modal features result in different text sequence dimensional features $T \in \{T, A\}$ and audio sequence dimensional features $A \in \{T, A\}$. Referring to the literature [36], we use 1D temporal convolutional layer as a sequence alignment tool to ensure that both are of the same dimension.

$$\hat{\mathbf{X}}_{T \in \{T,A\}}, \hat{\mathbf{X}}_{A \in \{T,A\}} = Conv1D(\{\mathbf{X}_T, \mathbf{X}_A\}, k\{T, A\}) \tag{16}$$

$k\{T, A\}$ means the size of the convolution kernel for text and audio modalities. $\hat{\mathbf{X}}_{T \in \{T,A\}}$ and $\hat{\mathbf{X}}_{A \in \{T,A\}}$ means the dimensionality data of text and audio features after convolution calculation.

Cross-modal fusion attention mechanism is one of the cores of AFR-BERT. Cross-modal Attention uses the information interaction between text and audio modalities to adjust the weights of the model and fine-tune the pre-trained language model BERT, as shown in Fig 3. $\hat{\mathbf{X}}_{T \in \{T,A\}}$ and $\hat{\mathbf{X}}_{A \in \{T,A\}}$ are the text features and audio features obtained from the data processing layer. The text interaction matrix N1 and audio interaction note matrix N2 are defined as

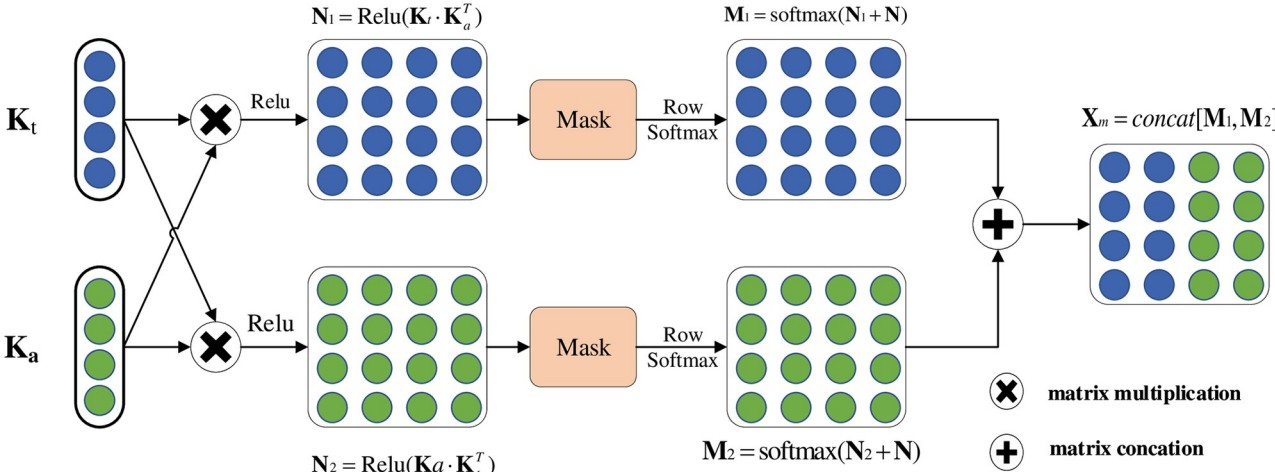

**Fig 3. Cross-modal fusion attention mechanism structure.** ($\mathbf{K}_t$) represents text feature data. ($\mathbf{K}_a$) represents audio feature data. (Relu, Row Softmax, softmax, concat) are all function calculations. (Mask) is a matrix.

below:

$$\mathbf{N}_1 = \text{Relu}(\hat{\mathbf{X}}_{T \in \{T,A\}} \cdot \hat{\mathbf{X}}_{A \in \{T,A\}}^T) \tag{17}$$

$$\mathbf{N}_2 = \text{Relu}(\hat{\mathbf{X}}_{A \in \{T,A\}} \cdot \hat{\mathbf{X}}_{T \in \{T,A\}}^T) \tag{18}$$

In the data processing layer, the model uses zero vector padding speech features with the same length as the text features. In order to reduce the influence of padding sequences, the model uses the mask matrix N in attention, after which the probability distribution of each feature sequence is calculated using soft attention to obtain a multimodal attention representation. 0 represents the location of the token. For the padding part, the feature data is calculated by the mask matrix as $-\infty$ (negative infinity), and the attention fraction of the filled position is 0 after the Softmax function calculation. Bimodal attention representation matrices M1 and M2 are defined as follows:

$$\mathbf{M}_1(i,j) = \frac{e^{\mathbf{N}_1(i,j) + \mathbf{N}(i,j)}}{\sum_{k=1}^{u} e^{\mathbf{N}_1(i,k) + \mathbf{N}(i,k)}} \text{ for } i, j = 1, 2..., u \tag{19}$$

$$\mathbf{M}_2(i,j) = \frac{e^{\mathbf{N}_2(i,j) + \mathbf{N}(i,j)}}{\sum_{k=1}^{u} e^{\mathbf{N}_2(i,k) + \mathbf{N}(i,k)}} \text{ for } i, j = 1, 2..., u \tag{20}$$

After obtaining the bimodal attention matrix, the feature representations of the two modalities are connected to help capture the important emotional factors between the multimodalities. We define the multimodal fusion matrix $\mathbf{X}_m$ as below:

$$\mathbf{X}m = concat[\mathbf{M}1, \mathbf{M}2] \tag{21}$$

**The Multimodal Association Layer** models the correlation of multimodal data.

The text modality emotional information is often closely related to the emotional changes of the audio modality. The emotional characteristics of audio patterns are usually related to frequency factors such as pitch, vocal intensity, loudness, and pitch length. How to filter a small amount of important information from a large amount of information, ensure that the reliance on external information is reduced, and capture important information with internal relevance is one of the cores of AFR-BERT research. In this paper, we adopt the self-attentive mechanism of transformer [10], also known as scaled dot product attention based on deflation. Its specific structure is shown in Fig 4, and the calculation expression is as follows:

$$Attention(Q, K, V) = SoftMax\left\{\frac{QK^T}{\sqrt{d_k}}\right\} \cdot V \tag{22}$$

We define multimodal data with computed internal correlations as $X_{Att}$. The expression is as follows:

$$X_{Att} = Attention(X_m) \tag{23}$$

The attention $X_{Att}$ and the last layer of BERT encoder text output sequence $R_t$ are processed with residual concatenation and normalization (Add&Norm). It makes the network effectively superimposed, avoids the degradation of network depth due to gradient disappearance, and also improves the accuracy and convergence speed of the model. The above calculation yields the classifiable aggregated total feature data $X_{classify}$, which is calculated by the following

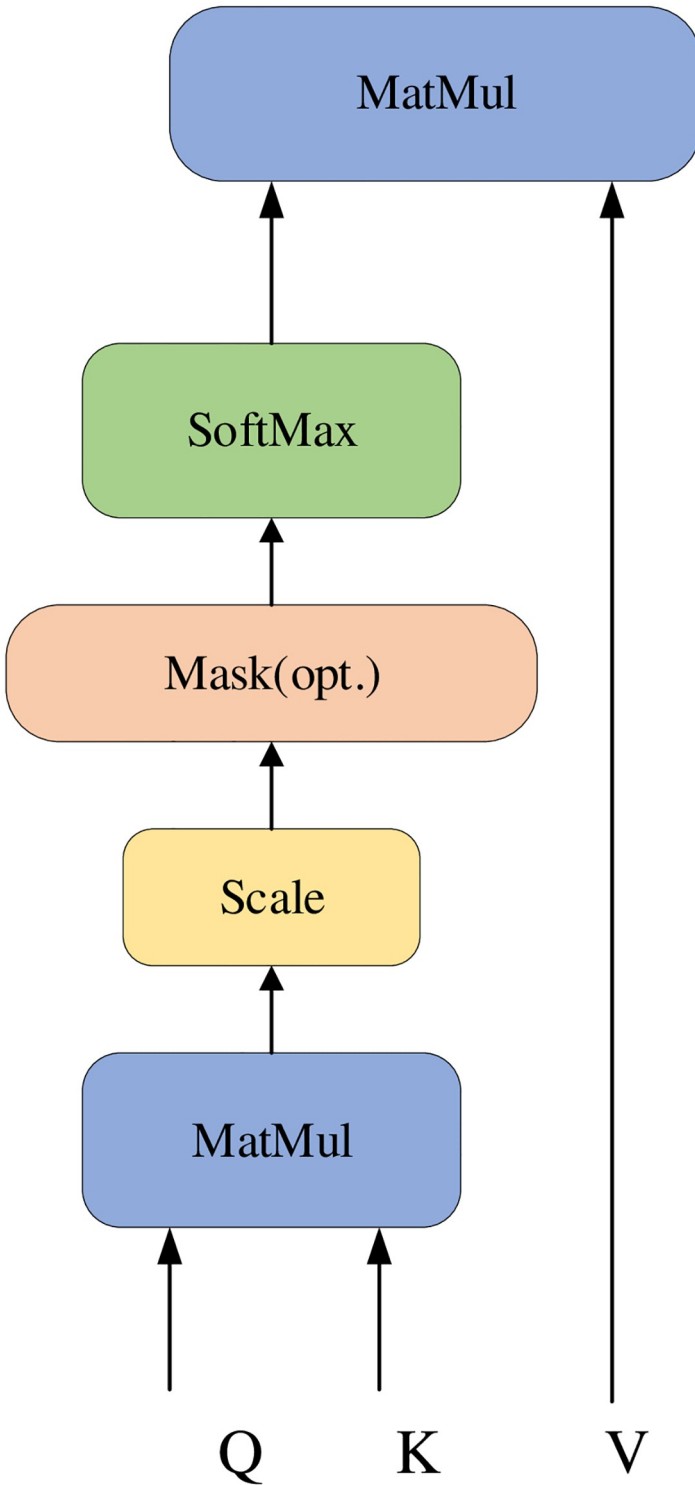

**Fig 4. Scaled dot product attention structure.** (Q) means the query matrix. (K) means the key matrix. (V) means the value matrix. (Mask) represents matrix operations for processing non-fixed-length sequences. ($\sqrt{d_k}$) is the scale factor for scaling.

**Table 3. The optimal parameter settings report.**

| Parameter | | | | | | |
|---|---|---|---|---|---|---|
| learning rate | batch size | max sequences | epoch | hidden dimensions of BiLSTM | loss function | optimizer |
| 0.0002 | 32 | 50 | 30 | 200 | MSE | Adam |

expression:

$$X_{\text{classify}} = Add\&Norm(X_{Att}, R_t) \tag{24}$$

**The Output Layer** outputs sentiment classification results.

The multimodal fused feature data $X_{classify}$ is computed by the fully connected layer and softmax function to derive the sentiment classification results with the following expressions:

$$y_i = soft \ \max(\mathbf{W}_{soft}(\tanh(\mathbf{W}_i\mathbf{X}_i + b_i) + b_{soft}) \tag{25}$$

$W_i$ and $b_i$ are the weight and bias of the fully connected. $W_{soft}$, and $b_{soft}$ are the weight and bias of the softmax layer. $X_i$ is the aggregated features. $y_i$ is the sentiment classification result.

## 3.5 Experimental settings

Parameters in deep learning can usually be divided into trainable parameters and hyperparameters. Trainable parameters can be optimally learned by backpropagation algorithms during model training, while hyperparameters are manually set to the correct values based on existing experience before training learning begins. The hyperparameters determine to some extent the final performance of the algorithm model. We use a basic Grid-Search to adjust the hyperparameters and select the best hyperparameter settings based on the performance of AFR-BERT on the validation set. For AFR-BERT, hyperparameters and tuning ranges are: learning rate (0.00001–0.001), batch size (16–128), max sequences length (32–96), the number of epochs (10–100), and hidden dimensions of BiLSTM (64–512). Mean Squared Error (MSE) is used as the loss function, and Adam is used as the optimizer.

Whenever the training of the AFR-BERT with a specific hyperparameter setting has finished, features learned from the AFR-BERT are used as input to the same downstream task models. The above optimal parameter settings report test result is shown in Table 3.

## 4. Results and discussion

We designed three sets of experiments to verify the validity of the model from different directions and discussed the experimental results through qualitative analysis.

## 4.1 Comparison experiments

A comparison experiment refers to setting up two or more experiments, followed by a comparative analysis of the experimental results to explore the relationship between various factors and the experimental subjects. We chose to conduct experiments comparing the baseline models and the AFR-BERT model on the CMU-MOSI and CMU-MOSEI.

Table 4 shows the experimental results of the evaluation metrics ($ACC_2$, $ACC_7$, F1, MAE) of the baseline models and AFR-BERT on the CMU-MOSI. The higher values of $ACC_2$, $ACC_7$, and F1 in the evaluation metrics prove the higher model performance, while the lower values of MAE prove the higher model performance.

From the experimental results in Table 4, it can be concluded that the AFR-BERT model produced new results on the CMU-MOSI and improved the related performance evaluation

**Table 4. Comparative experiments of multimodal sentiment analysis models on the dataset CMU-MOSI.**

| Model | MAE | ACC$_2$ | F1 | ACC$_7$ |
|:---:|:---:|:---:|:---:|:---:|
| TFN(B) | 0.901 | 80.82 | 80.77 | 34.94 |
| LMF(B) | 0.917 | 82.47 | 82.45 | 33.239 |
| MFN | 0.965 | 77.40 | 77.30 | - |
| RAVEN | 0.915 | 78.00 | 76.60 | - |
| ICCN | 0.860 | 83.00 | 83.00 | 39.00 |
| MFM(B) | 0.877 | 81.72 | 81.64 | 35.42 |
| MulT(B) | 0.861 | 83.00 | 82.80 | 40.00 |
| CM-BERT | 0.729 | 84.50 | 84.50 | **44.90** |
| MISA(B) | 0.783 | 83.40 | 83.60 | - |
| Self-MM(B) | 0.713 | 85.98 | 85.95 | - |
| MAG-BERT | 0.740 | 86.10 | 86.00 | - |
| AFR-BERT | **0.702** | **86.74** | **86.23** | 43.61 |

(B) means the language features are based on BERT. (-) means null value. The bolded values represent the best values of the performance indicators.

metrics. On the binary sentiment classification task, the AFR-BERT achieved 86.74% on ACC$_2$ with an improvement of 0.64%-9.34% compared to the baseline model. Similar to the result on ACC$_2$, our model achieved an improvement of 0.23%-9.63% on F1. In the sentiment score classification task, the AFR-BERT model achieved 43.61% on ACC$_7$, second only to CM-BERT. In the regression task, the AFR-BERT lowered the MAE value by about 0.26–0.11. What's more, most of the above baselines were analyzed using text, audio, and video, but our model created an outstanding result using only text and audio information.

To further capture the degree of correlation between the true values of sentiment and model predictions on the CMU-MOSI dataset, we used the pearson correlation coefficient (Corr) to measure the degree of linear correlation between the two values. Fig 5 shows the histogram of the comparative experimental results of the Corr index on the CMU-MOSI. By comparing the histograms, it was found that AFR-BERT and Self-MM achieved the optimal value of 0.798. The correlation between the model predicted values and the true labels approached a very strong degree of correlation. In the regression task, the AFR-BERT improved with an increase of 0.007–0.166 on Corr over other models except for Self-MM.

To demonstrate the generalizability of the AFR-BERT model, we conducted the same comparative experiments on the CMU-MOSEI. Table 5 shows the experimental results of the baseline model and AFR-BERT on the CMU-MOSEI to evaluate the metrics (ACC$_2$, F1, MAE). Since most of the baseline models did not produce experimental results of ACC$_7$ on the CMU-MOSEI, we do not compare the values of the metric.

From the experimental results in Table 5, it can be concluded that the AFR-BERT model produced better results on the CMU-MOSEI dataset and improved all performance evaluation metrics. On the binary sentiment classification task, the AFR-BERT model achieved 86.23% on ACC$_2$ with an improvement of 0.80% -10.23% compared to the baseline model. Similar to the result on ACC$_2$, our model achieved 86.15% on F1, which is an improvement of 0.85%-10.15% compared with baselines. In the regression task, the AFR-BERT model lowered the MAE value by about 0.006–0.093 compared to the baseline model.

To further capture the degree of correlation between the true values of sentiment and model predictions on the CMU-MOSEI, we used the pearson correlation coefficient (Corr) to

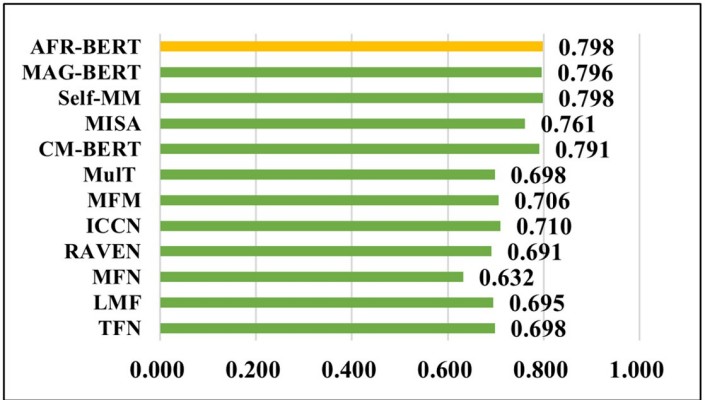

**Fig 5. Cross-sectional histograms of Corr metrics for each model on the CMU-MOSI.**

measure the degree of linear correlation between the two values. Fig 6 shows the histogram of the comparative experimental results of the Corr on the CMU-MOSEI dataset. By comparing the histogram, it was found that AFR-BERT and Self-MM achieved the optimal value of 0.772 close to 0.8. The correlation between the model predicted value and the true label is close to the degree of a very strong correlation. In the regression task, the AFR-BERT achieved an increase of 0.007–0.110 on Corr compared to the baseline model. Since the original papers of some models lack Corr data, it is empty.

It can be seen from the two sets of comparison experiments that the AFR-BERT model we proposed outperformed each classical multimodal sentiment analysis model in general in each performance metric, which fully verified the effectiveness and correctness of our method.

## 4.2 Ablation experiments

Ablation experiments are used to understand the network by removing parts of the network and studying the performance of the network for relatively complex neural networks. In order

**Table 5. Comparative experiments of multimodal sentiment analysis models on the dataset CMU-MOSEI.**

| Model | MAE | ACC$_2$ | F1 |
|---|---|---|---|
| TFN(B) | 0.901 | 80.82 | 80.77 |
| LMF(B) | 0.623 | 82.00 | 82.10 |
| MFN | - | 76.00 | 76.00 |
| RAVEN | 0.614 | 79.10 | 79.50 |
| ICCN(B) | 0.565 | 84.18 | 84.15 |
| MFM(B) | 0.568 | 84.40 | 84.30 |
| MulT (B) | 0.580 | 82.50 | 82.30 |
| MISA(B) | 0.555 | 85.50 | 85.30 |
| Self-MM(B) | 0.536 | 85.17 | 85.30 |
| MAG-BERT | - | 84.70 | 84.50 |
| AFR-BERT | **0.530** | **86.23** | **86.15** |

(B) means the language features are based on BERT. (-) means null value. The bolded values represent the best values of the performance indicators.

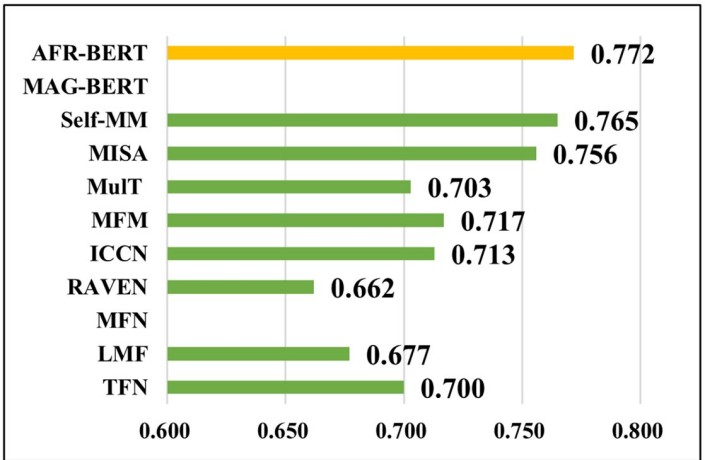

**Fig 6. Cross-sectional histograms of Corr metrics for each model on the CMU-MOSEI dataset.**

to investigate the effect of some modules on the performance of the overall model, four sets of ablation experiments were conducted on the CMU-MOSI.

**AFR-BERT(-BiLSTM).** We eliminated the BiLSTM module and inputted the extracted audio unimodal features and the text features output by encoder directly to the data fusion layer.

**AFR-BERT(-MFL).** We eliminated the multimodal fusion layer (MFL) and synthesized text features and audio features to transfer multimodal data to the data association layer.

**AFR-BERT(-MAL).** We eliminated the multimodal association layer (MAL) and inputted multimodal data directly to the output layer for sentiment classification. Table 6 shows our results of ablation experiments on the CMU-MOSI. We chose $ACC_2$, F1, and MAE as the model performance evaluation metrics.

From the experimental results, it can be seen that the AFR-BERT model shows a decreasing trend in each performance evaluation metric after removing the BiLSTM, MFL, and MAL modules, separately. After removing the BiLSTM module, firstly, $ACC_2$ decreased by 5.31%, and F1 decreased by 5.25% in the binary sentiment classification task, while MAE increased by 0.162 in the regression task. It shows that using BiLSTM to extract audio data can help the model to mine the emotional information in the data. After removing the MFL module, firstly, $ACC_2$ decreased by 8.2% and F1 dropped by 9.36% in the binary sentiment classification task, while the MAE increased by 0.242 in the regression task. It implies that the use of MFL with multimodal data can improve the effectiveness of the model for sentiment classification. After

**Table 6. Ablation experiments on the CMU-MOSI dataset.**

| Model | $ACC_2$ | F1 | MAE |
|---|---|---|---|
| ARF-BERT(-BiLSTM) | 81.43 | 80.98 | 0.864 |
| ARF-BERT(-MFL) | 78.54 | 76.84 | 0.944 |
| ARF-BERT(-MAL) | 80.47 | 79.43 | 0.907 |
| AFR-BERT | **86.74** | **86.23** | **0.702** |

(-) means subtracting the corresponding network. The bolded values represent the best values of the performance indicators.

removing the MAL module, first, $ACC_2$ decreased by 6.27% and F1 decreased by 6.8% in the binary sentiment classification task, whereas MAE increased by 0.205 in the regression task. It validates that MAL can filter out a small amount of important information from a large amount of information and effectively capture the internal correlation between multiple modalities. It can be seen from the ablation experimental results that BiLSTM has a great contribution in the feature extraction stage, MFL in the data fusion stage, and MAL in the data analysis stage.

## 4.3 Example analysis

To demonstrate the importance of multimodal data fusion and multimodal data correlation analysis, we selected some samples from the CMU-MOSI for testing. The sentiment polarity of each dialogue ranged between very strong negative (-3) and very strong positive (3). As shown in Table 7, each sample contained text and audio information, the true sentiment labels of the samples, and the predicted results of the AFR-BERT model. Examples 2 and 3 showed the presence of positive words such as "jokes", "laugh", "like" and "welcome" were easily judged as positive emotions from text modality alone, but their real emotions were nervous (negative) and low (negative). The model in this paper identified the correlation between text and audio data, mined the important features of multimodality, and accurately judged that the emotions of dialogues 3 and 4 were both negative, so the real emotions were predicted.

## 4.4 Qualitative analysis

Text modality has great limitations to understand the human emotions in Example 2 and Example 3 of Table 7, while AFR-BERT can adjust the emotional intensity by considering speech information to correctly capture human emotions. These examples show that AFR-BERT can better integrate text modality and audio modality to explore the hidden emotional information deeper.

In Table 6, we provide the results of the ablation experiments. It is clear from the experimental results that the modules do have a significant impact on the AFR-BERT performance. First, it shows that extracting audio data using BiLSTM can facilitate AFR-BERT to mine the emotional information in speech data. Second, it means that Multimodal Fusion Layer (MFL) can effectively fuse multimodal data and help improve the performance of AFR-BERT sentiment classification. Third, it shows that Multimodal Association Layer (MAL) can filter out a small amount of important information from a large number of messages and effectively

**Table 7. Sample analysis.**

| Case | | Data | True | AFR-BERT |
|---|---|---|---|---|
| 1 | Text | I really did enjoy as well wasn't too fond of the ending. | P(+1.4) | P(+0.6) |
| | Audio | Cheerful tone | | |
| 2 | Text | Except their eyes are kind of like this welcome to the polar express. | N(-0.6) | N(-0.2) |
| | Audio | Tense tone | | |
| 3 | Text | Maybe 5 jokes could make me laugh. | N(-1.8) | N(-0.3) |
| | Audio | Low tone | | |
| 4 | Text | But umm I liked it. | N(+1.8) | N(+0.5) |
| | Audio | Emphasis on tone | | |

(P) means positive emotions. (N) means negative emotions. The numbers represent emotional scores, where positive numbers represent positive scores and negative numbers represent negative scores.

capture the internal correlation between multiple patterns to help AFR-BERT discern the true sentiment information in the data.

Tables 4 and 5 show the results of the comparison experiments on the CMU-MOSI and CMU-MOSEI. From the experimental results, it is easy to see that the AFR-BERT model outperforms other baseline models in terms of performance evaluation metrics. This also confirms the correctness and high performance of the AFR-BERT model.

## 5. Conclusion

We propose an attention-based mechanism feature relevance fusion multimodal sentiment analysis model(ARF-BERT). The model is different from the traditional text unimodal sentiment analysis because we add audio modal information to obtain more comprehensive information and capture more sentiment features through the information interaction between text modality and audio modality. While we focused on multimodal data fusion, we reasonably analyzed the correlation between multimodal data, which greatly improved the effect of sentiment classification. Comparison experiments were conducted on CMU-MOSI and CMU-MOSEI. The experimental results prove that AFR-BERT can effectively improve the performance of multimodal sentiment analysis compared with the classical multimodal sentiment analysis model. Ablation experiments were conducted on the CMU-MOSI. The experimental results indicate that the model adopts BiLSTM, MFL, and MAL modules to contribute to the tasks of feature extraction, feature fusion, correlation extraction, and sentiment recognition of multimodal data. In future work, we will try to integrate visual information into text and audio information to dissect the sentiment changes more deeply and improve the performance of the model in sentiment analysis.

## Supporting information

**S1 Data. Processed dataset.** Processed dataset includes text and audio data.
(RAR)

## Author Contributions

**Conceptualization:** Zhou Jiawei.

**Data curation:** Zhou Jiawei.

**Formal analysis:** Ji Mingyu, Zhou Jiawei, Wei Ning.

**Funding acquisition:** Ji Mingyu.

**Investigation:** Zhou Jiawei, Wei Ning.

**Methodology:** Zhou Jiawei.

**Project administration:** Ji Mingyu, Zhou Jiawei.

**Resources:** Ji Mingyu.

**Software:** Zhou Jiawei.

**Supervision:** Ji Mingyu.

**Validation:** Ji Mingyu.

**Visualization:** Zhou Jiawei.

**Writing – original draft:** Zhou Jiawei.

**Writing – review & editing:** Ji Mingyu, Zhou Jiawei, Wei Ning.

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
