## [Decision Letter · Decision Letter 0]

2 Jun 2022

PONE-D-22-11384AFR-BERT: Attention-based mechanism feature relevance fusion multimodal sentiment analysis modelPLOS ONE

Dear Dr. Zhou,

Thank you for submitting your manuscript to PLOS ONE. After careful consideration, we feel that it has merit but does not fully meet PLOS ONE’s publication criteria as it currently stands. Therefore, we invite you to submit a revised version of the manuscript that addresses the points raised during the review process.

We look forward to receiving your revised manuscript.

Kind regards,

Sriparna Saha, PhD

Academic Editor

PLOS ONE

Journal Requirements:

"This work is sponsored by National Natural Science Foundation of China (Grant No.61901103).

We note that you have provided funding information. However, funding information should not appear in the Acknowledgments section or other areas of your manuscript. We will only publish funding information present in the Funding Statement section of the online submission form. 

"This work is sponsored by National Natural Science Foundation of China (Grant No. 435

61901103)."

Additional Editor Comments:

Both the reviewers have suggested some major changes; Authors are requested to incorporate the suggestions of both the reviewers to improve the paper.

Reviewers' comments:

Reviewer's Responses to Questions

**Comments to the Author**

1. Is the manuscript technically sound, and do the data support the conclusions?

Reviewer #1: Partly

Reviewer #2: Partly

2. Has the statistical analysis been performed appropriately and rigorously? 

Reviewer #1: Yes

Reviewer #2: No

3. Have the authors made all data underlying the findings in their manuscript fully available?

Reviewer #1: Yes

Reviewer #2: Yes

4. Is the manuscript presented in an intelligible fashion and written in standard English?

Reviewer #1: Yes

Reviewer #2: No

5. Review Comments to the Author

Reviewer #1: I am not sure because I am not find much difference between MISA and proposed methodology.

Kindly provide the Code.

Without experiment I am not able to comment anything.

Atleast provide some screen shots.

proofreading required.

Reviewer #2: Strengths:

1. Sections are detailed and informative for an average reader. All necessary information are provided.

2. The experiments performed are thorough with rigorous ablation studies.

Weakness:

1. The evaluation of the proposed method seems weak and more so, the improvement of the proposed method with the best performing baseline is marginal (in few decimal points), which limits the effectiveness of the approach.

2. One major problem I have with the manuscript, overall, is the poor writing style. There is huge scope for improvement in the presentation, specifically numerous typos, grammatical errors, etc. Some of the expositions are shown below. Considering myself as an average versed reader, it was very difficult to ignore all sort of typos (random capitalizations, missing punctuations, spellings, ) occurring in each and every line. The manuscript is not acceptable in its current form based on presentation issues alone.

Comments to the authors:

In table 6 caption, there is a mention of bolded values, however I don't find any.

At multiple places in the manuscript, the authors chose to write that their method 'significantly improves existing systems performances. This is slightly misleading as, (a). performance improvements are marginal compared to existing best method, (b). significance test not done to show that the obtained results are not attained by chance.

Typos and Expositions:

hard to follow. break into shorter sentences. "With the development of Internet technology and the popularity of many social media, in daily life, people often express their feelings about daily life and opinions on hot topics in microblogs, are used to reading product reviews before consumption, and like to write down their feelings after experiencing products and enjoying services, which consequently generates a large amount of multimodal data."

Wang [3]proposed a -> space should be given after each citation

Lin et al [4] used the -> al is followed by a fullstop

CMU-MOSI [12] (CMU Mul-timodal -> multimodal can be written without a hyphen or it may come after multi, but definitely after mul. same for 're-quire'

Sentiment Intensity)dataset -> remove paranthesis

NLP, BERT, etc. -> abbreviated terms must be defined with its full form at their first occurrence

Line 215, 'COVARER' - correct spelling

'method. a The 215 time step' -> misplaced 'a'

line 218, re-quires -> requires

line 418 -> ablation experiments: capitalization not done

In all equations, inconsistent superscript and subscript usage. line 232 mentioned X in caps which do not relate to any variable in the equations.

6. PLOS authors have the option to publish the peer review history of their article (what does this mean?). If published, this will include your full peer review and any attached files.

Reviewer #1: No

Reviewer #2: No

---

## [Author Response · Author response to Decision Letter 0]

8 Jul 2022

Dear Editors and Reviewers:

Thank you for your letter and for the reviewers' comments concerning our manuscript entitled “AFR-BERT: Attention-based mechanism feature relevance fusion multimodal sentiment analysis model” (PONE-D-22-11384). Those comments are all valuable and very helpful for revising and improving our paper, as well as the important guiding significance to our research. We have studied the comments carefully and have made corrections which we hope to meet with approval. Revised portions are marked in yellow on the paper. The corrections in the paper and the responses to the editors' and reviewers' comments are as follows: 

editors:

Response: Thank you very much for your review. After our tireless efforts, we have made corresponding changes to the style of the manuscript, the writing requirements, including the naming of the document. We hope our manuscript will meet your requirements, and we also hope to hear from you soon.

2. Please note that PLOS ONE has specific guidelines on code sharing for submissions in which author-generated code underpins the findings in the manuscript. In these cases, all author-generated code must be made available without restrictions upon publication of the work.

Response: We have placed the main code covered by this manuscript at https://github.com/Learn-Be/AFR-BERT.

3. We note that you have provided funding information. However, funding information should not appear in the Acknowledgments section or other areas of your manuscript. We will only publish funding information present in the Funding Statement section of the online submission form. Please remove any funding-related text from the manuscript and let us know how you would like to update your Funding Statement.

Response: We have removed any funding-related text from the manuscript.

Reviewers:

1. Is the manuscript technically sound, and do the data support the conclusions?

Reviewer #1: Partly

Reviewer #2: Partly

Response: Thank you very much for your constructive comments. To show that our results are not obtained by chance, we have added Section 3.5 (Experimental Settings) to our manuscript, which provides a detailed analysis of the experimental parameters of our method. At the same time, section 4.4 (Qualitative analysis) has been added to our manuscript to further explain and elaborate on the experimental history of the comparison, ablation and case studies. We sincerely hope that our modifications and improvements will meet with your approval.

2. Has the statistical analysis been performed appropriately and rigorously?

Reviewer #1: Yes

Reviewer #2: No

Response: Thank you very much for your constructive comments. In order to perform the statistical analysis correctly and rigorously, the analysis of the experimental results was fully refined and the results obtained for each experiment were synthesized in the qualitative analysis phase in Section 4.4 in order to obtain the best method.

3. Have the authors made all data underlying the findings in their manuscript fully available?

The PLOS Data policy requires authors to make all data underlying the findings described in their manuscript fully available without restriction, with rare exception (please refer to the Data Availability Statement in the manuscript PDF file). The data should be provided as part of the manuscript or its supporting information or deposited to a public repository. For example, in addition to summary statistics, the data points behind means, medians and variance measures should be available. If there are restrictions on publicly sharing data—e.g. participant privacy or use of data from a third party—those must be specified.

Reviewer #1: Yes

Reviewer #2: Yes

Response: Thank you very much for your affirmation and recognition of our work.

4. Is the manuscript presented in an intelligible fashion and written in standard English?

Reviewer #1: Yes

Reviewer #2: No

Response: We apologize for the poor language of our manuscript. We worked on the manuscript for a long time and the repeated addition and removal of sentences and sections obviously led to poor readability. We have now worked on both language and readability and have also involved native English speakers for language corrections. We really hope that the flow and language level have been substantially improved.

Reviewer 1:

1. I am not sure because I am not finding much difference between MISA and proposed methodology. Kindly provide the Code. Without experiment I am not able to comment anything. At least provide some screen shots. proofreading required.

Response: Thank you very much for your constructive comments. Your comments have had an unimaginable impact on the future direction of our team's research. We express our great appreciation. We find that MISA is fundamentally different from these existing works. MISA does not use contextual information and neither focus on complex fusion mechanisms. Instead, it stresses the importance of representation learning before fusion (MISA/2 RELATED WORK/ line 27-30). On the contrary, we have paid attention to the contextual information before fusion, proposed a new fusion algorithm for complex fusion mechanisms, and performed data analysis for the fused multimodal data. And we have elaborated and introduced the features of MISA method in our paper, which we hope can answer your questions (line 58-62). To refine the details of the experiment, we have added section 3.5 in our text, where we further describe the settings of the experimental parameters. We also place our method at https://github.com/Learn-Be/AFR-BERT.

Reviewer 2:

1. Sections are detailed and informative for an average reader. All necessary information is provided.

Response: Thank you very much for your affirmation.

2. The experiments performed are thorough with rigorous ablation studies.

Response: Thank you very much for your comments and for inspiring us.

3. The evaluation of the proposed method seems weak and more so, the improvement of the proposed method with the best performing baseline is marginal (in few decimal points), which limits the effectiveness of the approach.

Response: Thank you very much for your constructive comments. Your comments have helped us a lot in our research work. First of all, we acknowledge that the performance improvement of our proposed method is indeed limited compared to the best method, which is undeniable. However, it is well known that multimodal data is huge, complex and contains a large amount of interfering data, which makes the performance improvement very difficult. We also regret this. Second, our approach is not without merits. It contributes in the following ways.

 1. we get rid of the limitations of traditional unimodality and outperform the current optimal model using only text and audio data.

 2. Our approach takes a new approach to the research areas of feature extraction, multimodal data fusion and multimodal data analysis.

4. One major problem I have with the manuscript, overall, is the poor writing style. There is huge scope for improvement in the presentation, specifically numerous typos, grammatical errors, etc. Some of the expositions are shown below. Considering myself as an average versed reader, it was very difficult to ignore all sort of typos (random capitalizations, missing punctuations, spellings, ) occurring in each and every line. The manuscript is not acceptable in its current form based on presentation issues alone.

Response: We apologize for the poor language of our manuscript. We worked on the manuscript for a long time and the repeated addition and removal of sentences and sections obviously led to poor readability. We have now worked on both language and readability and have also involved native English speakers for language corrections. We really hope that the flow and language level have been substantially improved.

5. Comments to the authors:

In table 6 caption, there is a mention of bolded values, however I don't find any. At multiple places in the manuscript, the authors chose to write that their method 'significantly improves existing systems performances. This is slightly misleading as, (a). performance improvements are marginal compared to existing best method, (b). significance test not done to show that the obtained results are not attained by chance.

Response: We apologize for the writing errors in our manuscript. We are also very grateful to you for pointing out our mistakes. We have made detailed corrections and improvements to your comments, and we hope our efforts will meet your requirements. Details can be found in Table 7. We acknowledge that the performance improvement of our proposed method is indeed limited compared to the best method, which is undeniable. However, it is well known that multimodal data are huge, complex and contain a large amount of interfering data, which makes the performance improvement very difficult. We also regret this. To show that our results are not obtained by chance, we have added Section 3.5 (Experimental Settings) to our manuscript, which provides a detailed analysis of the experimental parameters of our method. At the same time, section 4.4 (Qualitative analysis) has been added to our manuscript to further explain and elaborate on the experimental history of the comparison, ablation and case studies.

6. Typos and Expositions:

a) hard to follow. break into shorter sentences. "With the development of Internet technology and the popularity of many social media, in daily life, people often express their feelings about daily life and opinions on hot topics in microblogs, are used to reading product reviews before consumption, and like to write down their feelings after experiencing products and enjoying services, which consequently generates a large amount of multimodal data."

Response: We are very sorry for the irregularities in writing. And thank you very much for pointing out where we went wrong. We have modified this long sentence. “With the development of internet technology, people often express their feelings about their daily lives and opinions on hot topics, are used to reading product reviews before consumption, and like to write down their feelings after experiencing products and enjoying services. This has led to the generation of a huge amount of multimodal data.”( line 2-5 )

b) Wang [3]proposed a -> space should be given after each citation.

Response: We are very sorry for the irregularities in writing. We have made a modification to the mistake.-> Wang et al. [3] proposed (line 11)

c) Lin et al [4] used the -> al is followed by a fullstop.

Response: We are very sorry for the irregularities in writing. We have made a modification to the mistake. -> Lin et al. [4] used the (line 12)

d) CMU-MOSI [12] (CMU Mul-timodal -> multimodal can be written without a hyphen or it may come after multi, but definitely after mul. same for 're-quire'

Response: We are very sorry for the irregularities in writing. We have made a modification to the mistake.-> CMU-MOSI (Multimodal Opinion Level Sentiment Intensity) (line 129)

e) Sentiment Intensity)dataset -> remove paranthesis.

Response: We are very sorry for the irregularities in writing. We have made a modification to the mistake.-> CMU-MOSI (Multimodal Opinion Level Sentiment Intensity) [12] is one (line 129)

f) NLP, BERT, etc. -> abbreviated terms must be defined with its full form at their first occurrence

Response: We are very sorry for the irregularities in writing. We have made a modification to the mistake.

g) Line 215, 'COVARER' - correct spelling. 

Response: We are very sorry for the irregularities in writing. We have made a modification to the mistake.-> COVAREP (line 215)

h) method. a The 215 time step' -> misplaced 'a'.

Response: We are very sorry for the irregularities in writing. We have made a modification to the mistake.-> The time step of (line 215)

i) line 218, re-quires -> requires.

Response: We are very sorry for the irregularities in writing. We have made a modification to the mistake.-> requires (line 217)

j) line 418 -> ablation experiments: capitalization not done.

Response: We are very sorry for the irregularities in writing. We have made a modification to the mistake.-> Ablation experiments (line 453)

k) In all equations, inconsistent superscript and subscript usage. line 232 mentioned X in caps which do not relate to any variable in the equations.

Response: We are very sorry for the irregularities in writing. We have made a modification to the mistake.-> xt is the input feature of the audio at moment t.(line 232)

We appreciate your warm work earnestly and hope that the corrections will meet with your approval. Once again, thank you very much for your comments and suggestions.

---

## [Decision Letter · Decision Letter 1]

18 Aug 2022

AFR-BERT: Attention-based mechanism feature relevance fusion multimodal sentiment analysis model

PONE-D-22-11384R1

Dear Dr. Zhou,

We’re pleased to inform you that your manuscript has been judged scientifically suitable for publication and will be formally accepted for publication once it meets all outstanding technical requirements.

Kind regards,

Sriparna Saha, PhD

Academic Editor

PLOS ONE

Additional Editor Comments (optional):

Reviewers' comments:

Reviewer's Responses to Questions

**Comments to the Author**

1. If the authors have adequately addressed your comments raised in a previous round of review and you feel that this manuscript is now acceptable for publication, you may indicate that here to bypass the “Comments to the Author” section, enter your conflict of interest statement in the “Confidential to Editor” section, and submit your "Accept" recommendation.

Reviewer #1: All comments have been addressed

Reviewer #2: All comments have been addressed

2. Is the manuscript technically sound, and do the data support the conclusions?

Reviewer #1: Partly

Reviewer #2: Yes

3. Has the statistical analysis been performed appropriately and rigorously? 

Reviewer #1: Yes

Reviewer #2: Yes

4. Have the authors made all data underlying the findings in their manuscript fully available?

Reviewer #1: Yes

Reviewer #2: Yes

5. Is the manuscript presented in an intelligible fashion and written in standard English?

Reviewer #1: Yes

Reviewer #2: Yes

6. Review Comments to the Author

Reviewer #1: Please Proofread the paper.

Please do t-test as results are too close.

Few lines have grammatical issues please solve the issue.

Reviewer #2: The authors have made sufficient revisions and have addressed all the concerns raised. Therefore, it could be accepted as a regular paper.

7. PLOS authors have the option to publish the peer review history of their article (what does this mean?). If published, this will include your full peer review and any attached files.

Reviewer #1: No

Reviewer #2: **Yes: **Soumitra Ghosh

---

## [Editor Report · Acceptance letter]

23 Aug 2022

PONE-D-22-11384R1 

AFR-BERT: Attention-based mechanism feature relevance fusion multimodal sentiment analysis model 

Dear Dr. Jiawei:

I'm pleased to inform you that your manuscript has been deemed suitable for publication in PLOS ONE. Congratulations! Your manuscript is now with our production department. 

Kind regards, 

on behalf of

Dr. Sriparna Saha 

Academic Editor

PLOS ONE